**Data Availability Statement:** The data are unavailable to the public because they contain personal identifying information and very sensitive

# Healthcare seeking behavior and delays in case of Drug-Resistant Tuberculosis patients in Bangladesh: Findings from a cross-sectional survey

Md. Zulqarnine Ibne Noman [1,2], Shariful Islam [2], Shaki Aktar [1,3], Ateeb Ahmad Parray [1,4]*, Dennis G. Amando [1], Jyoti Karki [1], Zafria Atsna [1], Dipak Kumar Mitra [5], Shaikh A. Shahed Hossain [1]

1 BRAC James P. Grant School of Public Health (JPGSPH), BRAC University, Dhaka, Bangladesh, 2 EcoHealth Alliance Bangladesh Programs, Institute of Epidemiology, Disease Control and Research (IEDCR), Dhaka, Bangladesh, 3 International Centre for Diarrhoeal Disease Research (icddr,b), Bangladesh, Dhaka, Bangladesh, 4 Department of International Health, Health Systems Program, The Johns Hopkins University, Baltimore, MD, United States of America, 5 Department of Public Health, North South University, Dhaka, Bangladesh

* ahmad.ateeb101@gmail.com

## Abstract

The emergence of Drug-Resistant Tuberculosis (DR-TB) has become a major threat globally and Bangladesh is no exception. Delays in healthcare seeking, proper diagnosis and initiation of treatment cause continuous transmission of the resistant tubercule bacilli through the communities. This study aimed to assess the different health care-seeking behaviors and delays among DR-TB patients in Bangladesh. A prospective cross-sectional study was conducted from November to December 2018, among 92 culture positive and registered DR-TB patients in four selected hospitals in Bangladesh. Data were collected through face-to-face interviews with survey questionnaire as well as record reviews. Among the 92 study participants, the median patient delay was 7 (IQR 3, 15) days, the median diagnostic delay was 88 (IQR 36.5, 210), the median treatment delay was 7 (IQR 4,12) days, and the median total delay among DR-TB patients was 108.5 (IQR 57.5, 238) days. 81.32% sought initial care from informal healthcare providers. The majority (68.48%) of the informal healthcare providers were drug sellers while 60.87% of patients sought care from more than four healthcare points before being diagnosed with DR-TB. The initial care seeking from multiple providers was associated with diagnostic and total delays. In Bangladesh, DR-TB cases usually seek care from multiple providers, particularly from informal providers, and among them, alarmingly higher healthcare-seeking related delays were noted. Immediate measures should be taken both at the health system levels and, in the community, to curb transmission and reduce the burden of the disease.

information about drug-resistant Tuberculosis patients in Bangladesh who are subjected to extreme stigma and discrimination. Ethical approval for the conduct of the study required that all data, including the locations of patients, be de-identified, imposed by Institutional Review Board (IRB) of BRAC James P Grant School of Public Health. For questions related to data availability please reach out to the IRB focal of BRAC JPGSPH here: irb-jpgsph@bracu.ac.bd.

**Funding:** The study was conducted under Summative Learning Project (SLP), as a partial requirement of Master of Public Health degree at BRAC James P Grant School of Public Health (BRAC JPGSPH), BRAC University. The SLP that this study is based on titled "Drug-resistance Tuberculosis in Bangladesh: epidemiological estimates, programmatic implications, and people's perspectives" was supported by Tropical Diseases Research (WHO-TDR), the Special Programme for Research and Training in Tropical Diseases, which is hosted at the World Health Organization and co-sponsored by UNICEF, UNDP, the World Bank and World Health Organization, under grant number: B40297 (Postgraduate training scheme: Building the capacity of the next generation of researchers and global health leaders, acquired by the researchers of BRAC JPGSPH). The funders had no role in study design, data collection and analysis, decision to publish, or preparation of the manuscript.

**Competing interests:** The authors have declared that no competing interests exist.

## Introduction

Tuberculosis (TB), caused by the Mycobacterium tuberculosis bacilli, ranks as the second deadliest infectious disease following the COVID-19 pandemic and is the 13th leading cause of death worldwide [1]. It's estimated that approximately one-quarter of the global population is infected with TB, yet only 10% of these individuals will develop clinical TB in their lifetime [1, 2]. During the 67th World Health Organization (WHO) Assembly in Geneva in 2014, a comprehensive strategy and targets, known as 'The End TB 2030,' were established. This strategy aims to reduce TB deaths by 90% and cut new cases by 80% [3]. However, the rise of drug-resistant tuberculosis (DR-TB) poses a significant challenge to the End TB-2030 goals. Currently, only one in four new DR-TB cases are detected, and merely half of these cases are successfully treated [3].

DR-TB refers to a clinical condition characterized by resistance to any of the first-line anti-TB drugs. It is classified into various forms: single-drug-resistant, multi-drug-resistant (MDR-TB), or extensive drug-resistant (XDR-TB) [4]. MDR-TB, the most prevalent form, shows resistance to both Rifampicin (RFP) and Isoniazid (INH), the primary first-line anti-TB drugs. XDR-TB, a more severe form, not only meets the criteria for MDR-TB but also exhibits resistance to any fluoroquinolone and at least one group-A second-line drug. XDR-TB, often referred to as 'super-bugs', poses a significant threat. If not effectively controlled and treated, it could lead to a global tuberculosis crisis [5, 6].

In 2020, approximately 157,000 incident cases of DR-TB were reported, and 150,000 new cases received treatment, as noted in the Global TB Report. However, it is estimated that each year there are around 500,000 new cases of DR-TB, but only a third of these receive treatment [7]. The COVID-19 pandemic significantly impacted DR-TB management, causing a 22% drop in new case detection and a 15% decrease in new treatment enrollments in 2019. This resulted in only one out of every three newly diagnosed DR-TB cases receiving treatment [1]. The challenge is compounded by cases that are undiagnosed, diagnosed late, or treated belatedly, leading to the continued spread of resistant bacilli within communities [8]. The high risk of transmission, elevated case fatality rate, prolonged treatment duration, and associated financial and social burdens make the control of DR-TB particularly challenging in low- and middle-income countries, such as Bangladesh [9]. Notably, Bangladesh is among the top 30 high TB burden countries. In 2020, there were 1,113 laboratory-confirmed new cases of DR-TB in Bangladesh, with an incidence rate of 2% as per the Global TB Report 2020 [9].

Timely diagnosis and initiation of treatment for DR-TB are crucial. Delays in starting treatment can lead to more severe complications, increased disease transmission, and higher mortality rates [10]. An untreated smear-positive TB case can infect up to 10 individuals annually and potentially over 20 throughout its natural course until death [11]. Healthcare-seeking behavior, which is the tendency of individuals to seek and use healthcare services, is influenced by a myriad of factors. These include socioeconomic conditions, health beliefs, stigma, accessibility and availability of health services, health literacy, trust in providers, peer influence, and social networks. Patient delays and diagnostic delays are closely linked to these patterns of care-seeking behavior. In South Asia, for instance, patients often first consult informal healthcare providers [12, 13]. In Bangladesh, drug sellers, village doctors, Ayurvedic practitioners, and homeopaths are commonly consulted as initial healthcare providers [14]. Notably, about 60.7% of patients in Bangladesh first seek care from non-qualified providers, predominantly drug sellers [15]. The path to diagnosis often involves visiting multiple care providers before reaching a DOTS center, the diagnostic point for DR-TB. The number of care-seeking points significantly influences the delay in diagnosis [16]. Moreover, initial suspicion of TB among healthcare providers is often low [17]. Furthermore, social stigma, fear of isolation, and

misconceptions contribute to delays in care-seeking, exacerbating patient delays [18]. Factors like gender, education, geographical location, and socioeconomic status also play a role in shaping DR-TB care-seeking behaviors in sub-Saharan Africa [19].

A study conducted in India identified a median patient delay of 25 days for those with uncomplicated pulmonary TB [20]. In Bangladesh, research from 2012–2013 revealed a median health system delay of 7.1 weeks and a median treatment initiation delay of 10 days among MDR-TB patients [21]. The World Health Organization (WHO) advises against any delay exceeding one day between screening to diagnosis and from diagnosis to the initiation of treatment, as per their Policy on TB infection control [22]. There is some debate in the literature about acceptable delay durations; some authors argue that a total delay of up to one month is permissible, while others suggest it should not surpass two months [23]. However, comprehensive studies examining the factors influencing healthcare-seeking behaviors and the various delays encountered by DR-TB patients, particularly in Bangladesh, are scarce.

The primary objective of this study is to evaluate the healthcare-seeking behaviors and associated delays in patients with drug-resistant tuberculosis (DR-TB). The insights gained from this research will be instrumental for TB control programs. They can utilize this evidence to devise specific measures at various levels of the health system, as well as within communities, aimed at minimizing these delays. Such strategic interventions are crucial for the successful implementation of the 'End TB Strategy' by 2030 [3].

## Methods

### Setting and study population

In this study, we employed a prospective cross-sectional design across Bangladesh to achieve our objectives, utilizing a quantitative survey approach. The healthcare landscape in Bangladesh for TB treatment includes 46 chest clinics, a dedicated 250-bed TB hospital, several medical college hospitals, and a specialized chest disease hospital. For our study, we purposefully selected four centers known for their high volume of TB and DR-TB cases, as indicated in prior research [24]. The study sites were the National Institute of Diseases of Chest and Hospital (NIDCH) in Dhaka, Mymensingh Tuberculosis and Leprosy Hospital, Netrakona Tuberculosis and Leprosy Hospital, and Tangail Tuberculosis and Leprosy Hospital (Fig 1). Additionally, to achieve our estimated sample size of 92, we interviewed 13 DR-TB patients in Upazilla health complexes (UHC) and surrounding communities in the catchment areas of the TB hospitals in Netrakona and Tangail. These community patients were identified from the MDR-TB register of the Damien Foundation (DF), an International Non-governmental Organization providing treatment to them. NIDCH, located in Mohakhali, Dhaka, is a top-tier referral hospital for DR-TB in Bangladesh, featuring 685 beds, with 70 dedicated to DR-TB patients [25]. The DF operates in 17 countries across Asia, including Bangladesh, where they collaborate with the National Tuberculosis Control Program (NTP) and manage three referral hospitals for MDR-TB treatment. These hospitals, located in Tangail, Mymensingh, and Netrakona, collectively offer 255 beds (Tangail 95, Mymensingh 100, and Netrakona 60) [26].

Between November 17th and December 7th, 2018, our study included all DR-TB patients admitted to the four selected TB hospitals, along with community patients from the nearby catchment areas of the TB hospitals in Tangail and Netrakona. The study population consisted of DR-TB patients who were bacteriologically confirmed, either via GeneXpert or culture testing. Our study focused on patients aged 16 or older, registered with the National Tuberculosis Control Program (NTP), encompassing all forms of DR-TB treatment registration groups. However, patients suffering from severe illnesses were excluded.

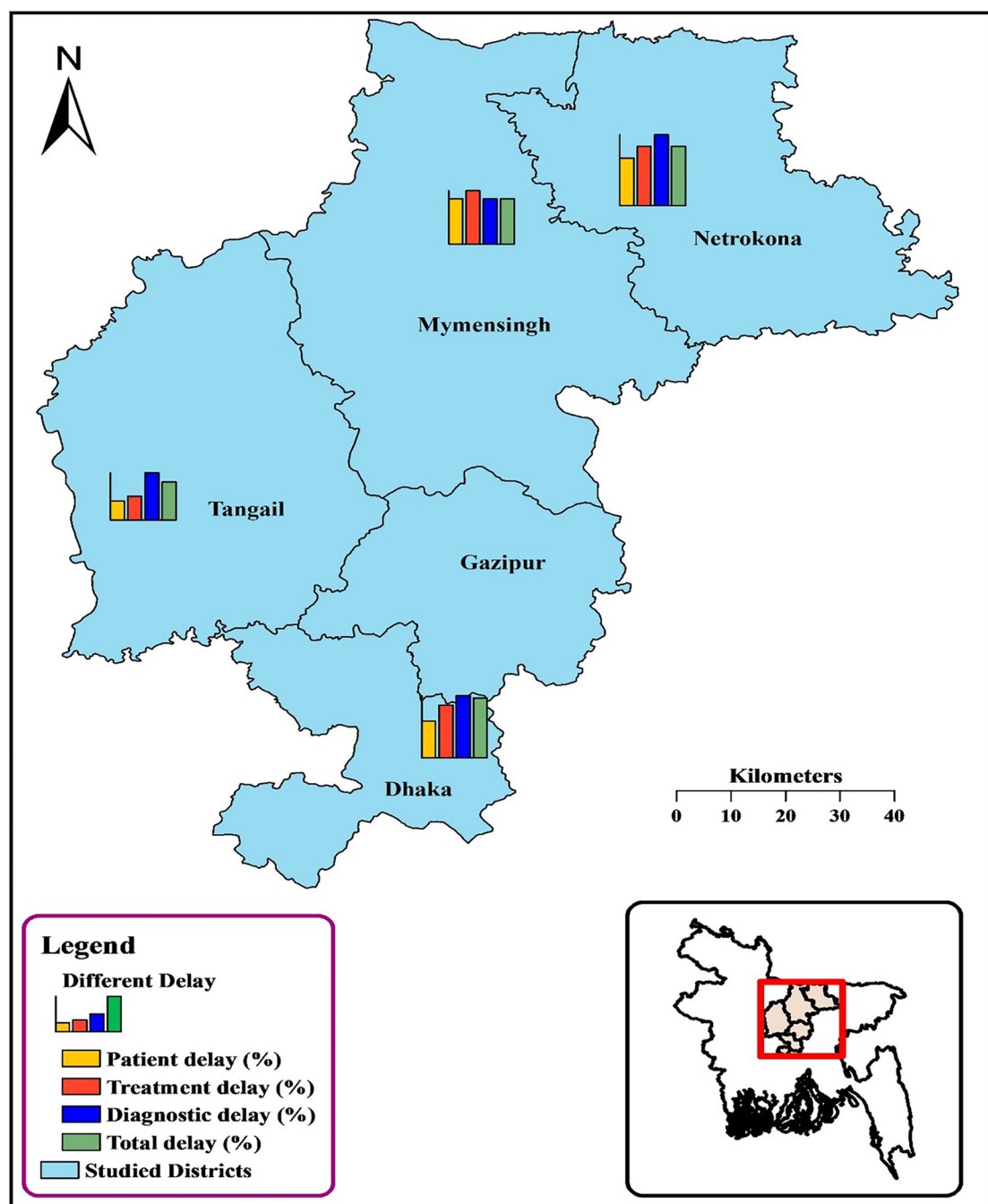

**Fig 1. Map of study site locations presenting the distribution of different delays among DR-TB patients in Bangladesh.** The map was created by utilizing a shapefile sourced from the freely accessible GADM database (GADM; www.gadm.org).

The distribution of participants across the study sites was as follows: 41 patients from the National Institute of Diseases of Chest and Hospital (NIDCH), 23 from the Mymensingh TB hospital, 9 from Tangail TB hospital, and 6 from Netrakona TB hospital. Additionally, 13 DR-TB patients from the catchment areas of Netrakona and Tangail TB hospitals were interviewed during the survey period.

## Study tools and data collection

For quantitative data collection, we utilized a pretested structured questionnaire. This tool was adapted from the World Health Organization's document 'Diagnostic and Treatment Delay in Tuberculosis—An In-Depth Analysis of the Health-Seeking Behavior of Patients and Health System Response in Seven Countries of the Eastern Mediterranean Region' [27]. To ensure linguistic accuracy and consistency, the questionnaire was translated into Bangla, the local language, and then back translated into English. Based on the insights gained from pretesting, the questionnaire underwent modifications as agreed upon by the research team.

Hospital records, including DR-TB cards, sputum registers, TB registers, and laboratory logs, were reviewed to corroborate the credentials of both hospital-admitted study participants and those from the catchment areas. The interviews were conducted by two researchers and three trained research assistants, all with medical backgrounds and proficiency in Bangla. Following each interview with a DR-TB patient, their TB card was examined to validate the information provided and to collect key data such as the treatment registration group, date of diagnosis, and treatment commencement. In cases of discrepancies between patient responses and record data, the latter was given precedence to minimize recall bias.

## Operational definitions

DR-TB is defined as resistance to one or more anti-TB drugs, and can be categorized as single-drug resistant, multi-drug resistant (MDR-TB), or extensive drug-resistant (XDR-TB). MDR-TB is specified as resistance to both first-line anti-TB drugs, Rifampicin (RFP) and Isoniazid (INH). Patients who had never received TB treatment before or had only taken anti-TB medications for less than one month were classified as new MDR-TB cases. Those who had previously been treated and declared cured or successfully completed treatment, but were later diagnosed with a recurrent episode, were categorized as relapse-MDR-TB. Patients who experienced treatment failure at the end of their most recent treatment course were classified as re-treatment MDR-TB. If the outcome of a patient's most recent TB treatment course was unknown or undocumented, they were categorized as other-MDR-TB [24]. XDR-TB was defined as resistance to first-line anti-TB drugs, one fluoroquinolone, and a second-line injectable drug (such as aminoglycosides or capreomycin).

The study assessed several timelines: the onset date of the first TB symptom, the date of initial care-seeking from any provider, the diagnosis date, and the treatment initiation date. Patient delay was the interval from when a patient first recognized TB symptoms to their initial consultation with any healthcare provider [20]. Diagnostic delay was the time from the first care-seeking to confirmed diagnosis of DR-TB via GeneXpert or culture [28]. Treatment delay referred to the period from DR-TB diagnosis to the start of treatment by healthcare providers [27]. The total delay was calculated as the duration from symptom recognition to treatment initiation [20]. Delays exceeding the median value were considered long delays (Fig 2).

Under the National Tuberculosis Control Program (NTP) guidelines in Bangladesh, DR-TB patients are classified into various treatment registration groups: new cases, relapse cases, failure of treatment with first-line drugs (F1), failure of treatment due to loss to follow-up, failure of retreatment with first-line drugs (F2), other cases, and XDR-TB. 'New cases' refers to patients with no previous history of TB. 'Relapse cases' are those who have been previously treated for TB, with treatment being completed and the patient cured. The 'other cases' category includes individuals with a history of TB but lacking documentation of drug intake or completion.

In this study, healthcare providers are differentiated as either formal or informal. Formal healthcare providers include public and private doctors, as well as providers within the DOTS

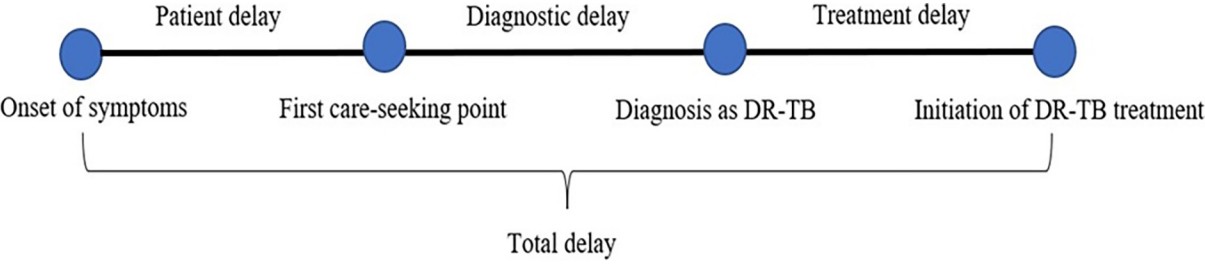

**Fig 2. Definition of different delays.**

(Directly Observed Treatment, Short course) system. Informal healthcare providers encompass drug sellers, village doctors (Locally Medically Trained, LMT), homeopaths, and Ayurvedic practitioners. Additionally, self-care at home is also considered a form of informal care.

## Data analysis

All collected data were entered and processed using MS Excel (Microsoft Office Professional Plus 2021, Microsoft Corporation, Washington, United States). The analysis was conducted using STATA version 16.0 (StataCorp, 4905 Lakeway Drive, College Station, Texas 77,845, USA). We performed a descriptive analysis of the socio-demographic characteristics, clinical, and personal profiles, using frequencies and percentages. Delays were quantified using median values and ranges.

For identifying predictive factors associated with different types of delays, we employed univariable logistic regression analysis. This facilitated the calculation of odds ratios (ORs) and 95% confidence intervals (CIs). Patient delays were estimated based on patient recall. Diagnostic delay was determined by deducting the patient delay from the diagnosis date, as recorded on the TB card. The treatment delay was calculated by subtracting the date of diagnosis from the date of treatment initiation, as per hospital records.

The map illustrating the study sites (Fig 1) was created using ArcGIS v10.8 (ESRI, Redlands, CA, United States). The shape files for this map were obtained from the GADM maps and data, a freely available resource (https://gadm.org/index.html) [29].

## Ethical approval

Ethical approval for this study was granted by the Institutional Review Board (IRB) of the James P. Grant School of Public Health, BRAC University, with the reference number 2018-40-IR. Additionally, formal permissions were obtained from the Damien Foundation Bangladesh's head office and the management of the respective hospitals involved in the study.

To ensure ethical compliance, informed written consent was obtained from all respondents aged eighteen and above, following a thorough verbal explanation of the study's purpose and procedures. For participants under eighteen, written consent was secured from their parents or guardians, along with the assent of the minors themselves. In cases where participants had not received formal education or expressed a preference for an alternative method, thumbprints were collected as consent.

The anonymity and confidentiality of all respondents were rigorously maintained throughout the study. To ensure safety during interviews, Personal Protective Equipment (PPE), including N95 masks, was utilized. Interviews with community patients were conducted at the nearest Directly Observed Treatment, Short-course (DOTS) centers. As a token of

appreciation and to facilitate participation, transport costs were reimbursed to community participants.

## Result

In this study, we interviewed a total of 92 patients. Of these, 25 (27.17%) were categorized as new MDR-TB cases, 29 (31.52%) as Relapse-MDR-TB, 33 (35.87%) as Re-treatment-MDR-TB, 3 (3.26%) as XDR-TB, and 2 (2.17%) fell into the category of other MDR-TB patients. Demographically, a majority of the participants, 69 (75%), were male. The age group most represented was 16–34 years, accounting for 45 (48.91%) patients, and a significant proportion, 66 (71.74%), resided in rural areas. The majority of participants were married, 76 (82.61%), and the largest occupational group was manual workers, representing 40 (43.48%) of the patients. Educationally, 42 (45.65%) of the patients had no formal schooling. Regarding smoking habits, more than half of the participants, 47 (51.09%), were former smokers, and 15 (16.30%) were current smokers. In terms of economic status, the largest group, comprising 38 (41.30%) of the participants, had a family income ranging between 10,001 to 20,000 Bangladeshi Taka (BDT).

A subset of the participants, 6 (6.52%), were identified as alcoholics. Notably, a considerable number of patients had comorbid conditions: 16 (17.39%) were diagnosed with diabetes, and 10 (10.87%) had other co-morbid diseases. Furthermore, a history of previous TB infection was reported in 62 (67.39%) of the patients. Additionally, evidence of BCG vaccination, indicated by the presence of a BCG scar, was observed in 60 (65.22%) of the participants.

The predominant symptoms reported by patients at their initial healthcare consultation included fever in 83 (90.22%) cases and cough in 78 (84.87%). Other common symptoms were weight loss in 61 (66.30%) patients, cough with blood in 12 (13.04%), chest pain in 23 (25%), and loss of appetite in 10 (10.87%). Notably, a majority of the patients, 67 (72.83%), presented with three or fewer symptoms initially. When seeking their first medical consultation, most patients, 74 (81.32%), chose informal healthcare providers. Additionally, a significant proportion of the patients, 56 (60.87%), consulted more than four different healthcare points before receiving a confirmatory diagnosis (Table 1).

The study revealed that the median patient delay was 7 days, with an interquartile range (IQR) of 3 to 15 days. The median diagnostic delay was notably longer, at 88 days (IQR 36.5 to 210 days), and the median treatment delay was found to be 7 days (IQR 4 to 12 days). Consequently, the total delay amounted to a median of 108.5 days (IQR 57.5 to 238 days) (Fig 3). Additionally, patients consulted a median of 5 different healthcare points (IQR 4 to 6) before receiving a diagnosis. In terms of mean values, the patient delay was 15.38 days, the diagnostic delay was 123.46 days, the treatment delay was 9.63 days, and the mean total delay amounted to 148 days.

More than half of the DR-TB patients, 54.35%, experienced a longer total delay. Notably, the highest proportions of extended total delays were observed in patients from NIDCH (63.41%) and Netrakona (62.50%). In Netrakona, half of the participants (50.00%) experienced a longer patient delay. Additionally, a significant majority in Netrakona faced extended diagnostic delays (75.00%) and treatment delays (62.50%) (Map-1, Table 2).

The median patient delay exhibited notable variations across different demographic and socioeconomic categories. Men experienced a considerably longer median patient delay of 14 days (IQR 3 to 25 days), nearly double that of women, who had a median delay of 6 days (IQR 3 to 10 days). Among different age groups, the young adult cohort (16–34 years) faced a longer median patient delay of 7 days (IQR 3 to 15 days) compared to the 35–54 years age group (6.5 days) and those aged 55 years and above (5.5 days). Ever-married patients had a higher median patient delay of 7 days (IQR 3 to 15 days) than the never-married group, which had a median

**Table 1. Frequency distribution with sociodemographic factors, clinical symptoms, care seeking variables.**

| Characteristics | Number of patients (%) |
|---|---|
| **Sex** | |
| Men | 69 (75) |
| Women | 23 (25) |
| **Age (year)** | |
| 16–34 | 45 (48.91) |
| 35–54 | 23 (25.00) |
| ≥55 | 24 (26.09) |
| **Residence** | |
| Rural | 66 (71.74) |
| Urban | 22 (23.91) |
| Semi-urban | 04 (4.35) |
| **Marital Status** | |
| Ever married | 76 (82.61) |
| Never married | 16 (17.39) |
| **Employment** | |
| Unemployed | 9 (9.78) |
| Sales & service | 16 (17.39) |
| Business | 10 (10.87) |
| Manual work | 40 (43.48) |
| Housewife | 9 (9.78) |
| Other | 8 (8.70) |
| **Educational status** | |
| Not attended school | 42 (45.65) |
| Primary | 19 (20.65) |
| Secondary & above | 31 (33.70) |
| **Current smoker** | |
| Yes | 15 (16.30) |
| No | 77 (83.70) |
| **Past smoker** | |
| Yes | 47 (51.09) |
| No | 45 (48.91) |
| **Family income (BDT)** | |
| ≤10000 | 32 (34.78) |
| 10001–20000 | 38 (41.30) |
| ≥20000 | 22 (23.91) |
| **Consume alcohol** | |
| Yes | 6 (6.52) |
| No | 86 (93.48) |
| **DM** | |
| Yes | 16 (17.39) |
| No | 76 (82.61) |
| **BCG scar** | |
| Yes | 60 (65.22) |
| No | 21 (22.83) |
| Don't know | 11 (11.96) |
| **History of the previous TB** | |
| Yes | 62 (67.39) |
| No | 30 (32.61) |

(*Continued*)

**Table 1.** (Continued)

| Characteristics | Number of patients (%) |
|---|---|
| **Number of care-seeking points** | |
| ≤5 | 53 (57.61) |
| >5 | 39 (42.39) |
| **Co-morbid disease** | |
| Yes | 10 (10.87) |
| No | 82 (89.13) |
| **Treatment registration group** | |
| New, MDR | 25 (27.17) |
| Relapse, MDR | 29 (31.52) |
| Re-treatment, MDR | 33 (35.87) |
| Other, MDR | 2 (2.17) |
| X-DR | 3 (3.26) |
| **Symptoms** | |
| Fever | 83 (90.22) |
| Cough | 78 (84.78) |
| Weight loss | 61 (66.30) |
| Chest pain | 23 (25.00) |
| Cough with blood | 12 (13.04) |
| Loss of appetite | 10 (10.87) |
| Other | 04 (4.35) |
| **Number of initial symptoms** | |
| ≤3 | 67 (72.83) |
| >3 | 25 (27.17) |
| **Type of first healthcare provider** | |
| Formal | 17 (18.68) |
| Informal | 74 (81.32) |
| **Number of care-seeking points** | |
| ≤4 | 36 (39.13) |
| >4 | 56 (60.87) |

of 5 days (IQR 3 to 14.5 days). Unemployed individuals reported a median patient delay of 14 days (IQR 3 to 15 days), contrasting with other occupational groups such as manual workers, housewives, and businessmen. Notably, those engaged in sales and service occupations had a shorter median patient delay of 3.5 days (IQR 3 to 7.5 days). Patients with a monthly family income of ≥20,000 BDT experienced a median patient delay of 4 days (IQR 3 to 7 days), while those earning less than 20,000 BDT per month faced a longer delay of 7 days (IQR 3 to 15 days). Additionally, patients with comorbid diseases tended to have a shorter median patient delay of 4.5 days (IQR 3 to 7 days) compared to those without comorbid conditions, who had a median delay of 7 days (IQR 3 to 15 days) (Table 3).

The study found that the median diagnostic delay differed significantly across various groups. Women experienced a longer median diagnostic delay of 102 days (IQR 33 to 180 days) compared to men, who had a median delay of 87 days (IQR 39 to 224 days). Young adults (16–34 years) faced a median diagnostic delay of 101.5 days (IQR 36 to 201 days), which was longer than other age groups. Urban residents had a shorter median diagnostic delay of 60 days (IQR 31 to 129 days) compared to their rural counterparts, who experienced a delay of 90 days (IQR 37 to 226 days).

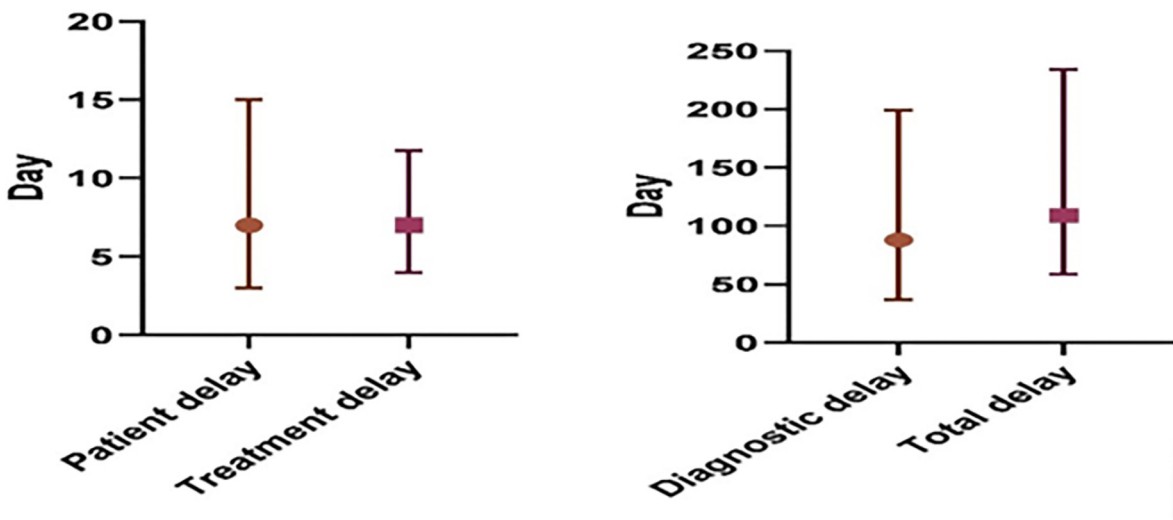

**Fig 3. Median and IQR of different delays among DR-TB patients in Bangladesh.**

Never-married patients encountered a median diagnostic delay of 125 days (IQR 23.5 to 211.5 days), which was longer than that of ever-married individuals, who had a delay of 84 days (IQR 42 to 206.5 days). Housewives reported a median diagnostic delay of 106 days (IQR 62 to 142 days), higher than other professional groups. Patients with higher education levels, up to secondary and beyond, had a median diagnostic delay of 81 days (IQR 36 to 180 days).

Current smokers experienced a longer delay of 125 days (IQR 60 to 237 days) compared to non-smokers, who had a delay of 84 days (IQR 36 to 194 days). Patients with lower incomes faced a median diagnostic delay of 103 days (IQR 56 to 226 days), more than those with higher incomes. Diabetic patients had a shorter diagnostic delay of 32 days (IQR 23 to 64 days) compared to non-diabetics, who had a delay of 98 days (IQR 60 to 222 days).

Patients with a history of previous TB infection had a median diagnostic delay of 105 days (IQR 62 to 231 days), which was longer than those without a previous TB history, who experienced a delay of 39 days (IQR 24 to 89 days). A longer median diagnostic delay of 142 days (IQR 72 to 237 days) was observed among patients who consulted more than four care-seeking points, compared to a delay of 62 days (IQR 30 to 134 days) for those who consulted four or fewer points.

Patients initially seeking care from formal healthcare providers had a shorter median diagnostic delay of 39 days (IQR 23 to 106 days) than those who consulted informal providers,

**Table 2. Percentage of DR-TB patients who had long delays according to the study sites*.**

| Long delays | NIDCH (%) n = 41 | Mymensingh (%) n = 23 | Tangail (%) n = 20 | Netrakona (%) n = 8 | Overall (%) n = 92 |
|---|---|---|---|---|---|
| Patient delay | 39.02 | 47.83 | 20.00 | 50.00 | 38.04 |
| Diagnostic delay | 65.85 | 47.83 | 50.00 | 75.00 | 52.17 |
| Treatment delay | 56.10 | 56.52 | 25.00 | 62.50 | 50.00 |
| Total delay | 63.41 | 47.83 | 40.00 | 62.50 | 54.35 |

* The table shows the percentages of DR-TB patients who had long patients, diagnostic, treatment, and total delay according to the study sites. Patient delay > 7 days, diagnostic delay > 88 days, treatment delay > 7 days, and total delay > 108 days are considered as long delay.

who faced a delay of 95 days (IQR 52 to 226 days). Patients with co-morbid diseases experienced a median diagnostic delay of 67 days (IQR 33 to 98 days), which was shorter than those without co-morbid conditions, who had a delay of 90 days (IQR 37 to 219 days). Patients presenting with three or fewer initial symptoms had a median diagnostic delay of 80 days (IQR 32.5 to 174.5 days) (Table 3).

The study observed that the median treatment delay varied across different groups. Women, individuals aged 55 years and above, rural residents, those who had never attended school, patients with a history of previous TB infection, manual workers, and those in 'Sales & Service' occupations experienced longer median treatment delays of 8 to 8.5 days. Conversely, patients who initially sought care from formal healthcare providers and those with co-morbid diseases had shorter median treatment delays of 5.5 and 6 days, respectively (Table 3).

Regarding the median total delay, longer durations were observed among women (124 days), rural patients (108 days), never-married individuals (141.5 days), housewives (134.5 days), manual workers (104 days), those with a family income of ≤10,000 BDT (129 days), patients with a history of previous TB (119 days), those who consulted more than four care-seeking points (176 days), and patients presenting with more than three initial symptoms (150 days). On the contrary, shorter median total delays were found among patients aged 55 years and above (87 days), urban residents (68 days), diabetic patients (63 days), those who first sought care from formal healthcare providers (67.5 days), and patients with co-morbid diseases (75 days) (Table 3).

In the logistic regression analysis, certain factors were found to be significantly associated with different types of delays. Patient delay was significantly associated with being never married (Odds Ratio [OR] = 4.57, 95% Confidence Interval [CI] = 1.20–17.33) and being non-diabetic (OR = 3.19, 95% CI = 1.01–10.10) (Table 4). The diagnostic delay showed a significant association with having consulted more than four care-seeking points (OR = 3.33, 95% CI = 1.38–8.03) and being non-diabetic (OR = 10.16, 95% CI = 2.15–47.90).

No variables were found to be significantly associated with treatment delay. However, total delay was significantly associated with being non-diabetic (OR = 5.65, 95% CI = 1.49–21.45) and consulting more than four care-seeking points (OR = 3.79, 95% CI = 1.55–9.24).

## Discussion

The healthcare-seeking behaviors of DR-TB patients critically influence their disease progression, prognosis, and eventual outcomes. The choice and sequence of healthcare providers play a pivotal role in the path to recovery. In Bangladesh, a pluralistic health system prevails, marked by the coexistence of various stakeholders, including both formal and informal healthcare providers [30]. However, the referral system, particularly from informal to formal providers, is not well-established. This study found that the majority of DR-TB patients initially consulted informal providers, predominantly drug sellers. This aligns with findings from other studies on TB/MDR-TB care-seeking behaviors in Bangladesh [14, 31].

Comparable patterns were observed in studies conducted among TB patients in India, Zimbabwe, and Ethiopia, where two-thirds initially sought care from non-qualified professionals [32–34]. Often, these informal providers treat patients with medications without recommending any diagnostic investigations [15] and do not refer patients to formal healthcare providers or DOTS centers. Consequently, our study noted significant diagnostic delays among patients who first sought care from informal providers. These patients experienced more than double the median diagnostic delay compared to those who initially consulted formal providers. A previous study also reported longer health system delays in TB patients who sought care from informal providers (52%) as opposed to formal providers (16%) [35].

**Table 3. Socio-demographic characteristics and personal history of DR-TB patients.**

| Characteristics | N (%) | Patient delay | Diagnosis delay | Treatment initiation delay | Total Delay |
|---|---|---|---|---|---|
| | | | Median delay (IQR) in days* | | |
| **Sex** | | | | | |
| Men | 69 (75) | 6 (3–10) | 87 (39–224) | 6 (4–9) | 104 (48–241) |
| Woen | 23 (25) | 14 (3–25) | 102 (33–180) | 8 (4–14) | 124 (57.5–213.5) |
| **Age (year)** | | | | | |
| 16–34 | 45 (48.91) | 7 (3–15) | 101.5 (36–201) | 7 (4–11) | 119 (55–240) |
| 35–54 | 23 (25.00) | 6.5 (3–15) | 96.5 (64–184) | 7 (4–14) | 115 (102–190) |
| ≥55 | 24 (26.09) | 5.5 (1.5–11) | 64.5 (32–221.5) | 8 (5–13.5) | 87 (44–235.5) |
| **Residence** | | | | | |
| Rural | 66 (71.74) | 7 (3–14) | 90 (37–226) | 8 (5–13) | 108 (61–250) |
| Urban | 22 (23.91) | 7 (3–30) | 60 (31–129) | 6 (3.5–12) | 68 (45–162) |
| Semi-urban | 04 (4.35) | 3 (1–6) | 140.5 (84.5–207.5) | 4.5 (3.5–8) | 149.5 (95–215.5) |
| **Marital Status** | | | | | |
| Ever married | 76 (82.61) | 7 (3–15) | 84 (42–206.5) | 7 (4–11) | 106 (62–234.5) |
| Never married | 16 (17.39) | 5 (3–14.5) | 125 (23.5–211.5) | 8 (3.5–13.5) | 141.5 (56–238.5) |
| **Employment** | | | | | |
| Unemployed | 9 (9.78) | 14 (3–15) | 52 (37–201) | 6 (4–8) | 61 (55–241) |
| Sales & service | 16 (17.39) | 3.5 (3–7.5) | 80 (24–222) | 8.5 (4–12.5) | 90 (42–236) |
| Business | 10 (10.87) | 7 (3–15) | 64 (30–81) | 7 (4–9) | 91 (47–120) |
| Manual work | 40 (43.48) | 7 (3–10) | 89.5 (60–224) | 8 (4–17) | 104 (78–240) |
| Housewife | 9 (9.78) | 7 (1.5–20) | 106 (62–142) | 6 (5–8) | 134.5 (88.5–189.5) |
| Other | 8 (8.70) | 18 (6–30.5) | 137 (56.5–255) | 4 (3–8) | 208.5 (68–296) |
| **Educational status** | | | | | |
| Not attended school | 42 (45.65) | 6 (2–14) | 87 (45–219) | 8 (5–13) | 103 (66.5–241) |
| Primary | 19 (20.65) | 7 (3–15) | 104 (60–222) | 7 (3–13) | 115 (81–162) |
| Secondary & above | 31 (33.70) | 7 (3–15) | 81 (36–180) | 7 (3–11) | 102 (48–241) |
| **Current smoker** | | | | | |
| Yes | 15 (16.30) | 7 (4–15) | 125 (60–237) | 7.5 (5–11) | 119 (78–255) |
| No | 77 (83.70) | 7 (3–14.5) | 84 (36–194) | 7 (4–13) | 104 (55–229) |
| **Past smoker** | | | | | |
| Yes | 47 (51.09) | 7 (2–10) | 91 (37–231) | 8 (4–14) | 103 (54.5–246) |
| No | 45 (48.91) | 7 (3–15) | 85 (36–180) | 7 (4–10.5) | 111.5 (57.5–213.5) |
| **Family income (BDT)** | | | | | |
| ≤10000 | 32 (34.78) | 7 (3–15) | 103 (56–226) | 7 (4–9) | 129 (81–242) |
| 10001–20000 | 38 (41.30) | 7 (3–15) | 60 (19–180) | 8 (3.5–13) | 82 (41–195) |
| ≥20000 | 22 (23.91) | 4 (3–7) | 96 (62–194) | 7 (5–14) | 106.5 (69– |
| **Consume alcohol** | | | | | |
| Yes | 6 (6.52) | 3.5 (3–6) | 258.5 (95–297) | 7 (4–7) | 236 (104–303) |
| No | 86 (93.48) | 7 (3–15) | 81 (33–184) | 7.5 (4–13) | 108 (55–229) |
| **DM** | | | | | |
| Yes | 16 (17.39) | 7 (3.5–20) | 32 (23–64) | 5.5 (3–9) | 63 (33–115) |
| No | 76 (82.61) | 7 (3–15) | 98 (60–222) | 8 (4–13) | 112 (78–241) |
| **BCG scar** | | | | | |
| Yes | 60 (65.22) | 7 (2–10) | 80.5 (37–180) | 7 (3–11) | 102 (58–224) |
| No | 21 (22.83) | 7 (3–15) | 97.5 (21–201.5) | 7.5 (5.5–18.5) | 112 (33–229) |
| Don't know | 11 (11.96) | | 180 (60–295) | 7.5 (4–12) | 201.5 (93–291.5) |
| **History of the previous TB** | | | | | |

*(Continued)*

**Table 3.** (Continued)

| Characteristics | N (%) | Patient delay | Diagnosis delay | Treatment initiation delay | Total Delay |
|---|---|---|---|---|---|
| Yes | 62 (67.39) | 7 (3–12) | 105 (62–231) | 8 (4.5–11) | 119 (84–252) |
| No | 30 (32.61) | 7 (3–21) | 39 (24–89) | 6 (4–14) | 64.5 (42.5–145) |
| **Number of care-seeking points** | | | | | |
| ≤4 | 53 (57.61) | 7 (3–15) | 62 (30–134) | 7 (4–10.5) | 78 (47–150) |
| >4 | 39 (42.39) | 7 (3–15) | 142 (72–237) | 8 (4–16) | 176 (90–255) |
| **Number of initial symptoms** | | | | | |
| ≤3 | 67 (72.83) | 7 (3–15) | 80 (32.5–174.5) | 8 (5–13) | 108 (58–224) |
| >3 | 25 (27.17) | 6 (3–10) | 143.5(53.5–239.5) | 7 (3.5–8.5) | 150 (55–242) |
| **Type of first healthcare provider** | | | | | |
| Formal | 17 (18.68) | 8 (3–30) | 39 (23–106) | 5.5 (2.5–10.5) | 67.5 (34–145) |
| Informal | 74 (81.32) | 7 (3–14) | 95 (52–226) | 8 (4–13.5) | 109 (63–250) |
| **Co-morbid disease** | | | | | |
| Yes | 10 (10.87) | 4.5 (3–7) | 67 (33–98) | 6 (4–9) | 75 (55–98) |
| No | 82 (89.13) | 7 (3–15) | 90 (37–219) | 7.5 (4–12.5) | 111.5 (57.5–240.5) |
| **Treatment registration group** | | | | | |
| New, MDR | 25 (27.17) | 7 (3–15) | 36.5 (23–109) | 5 (4–13) | 57.5 (37.5–145) |
| Relapse, MDR | 29 (31.52) | 7 (3–10) | 92.5 (30–238.5) | 7.5 (4–11.5) | 102 (47–255) |
| **Re-treatment, MDR** | 33 (35.87) | 5 (3–15) | 105 (70–194) | 8 (5.5–11) | 119 (90–236) |
| Other, MDR | 2 (2.17) | 10.5 (7–14) | 119 (37–201) | 6.5 (4–9) | 136 (48–224) |
| X-DR | 3 (3.26) | 21 (15–30) | 81 (64–283) | 19.5 (18–21) | 215.5 (115–316) |

To effectively control DR-TB, policymakers should consider the significant role of informal caregivers. Implementing awareness programs, training these providers, and establishing a structured referral system from informal to formal caregivers could be pivotal in reducing diagnostic and treatment delays.

This study assessed the number of care-seeking points from the initial symptoms of DR-TB to the confirmation of diagnosis. It was found that the median number of care-seeking points for DR-TB patients in Bangladesh is notably higher compared to other countries, such as the USA (4), India (2), and Zimbabwe (3) [33, 34, 36]. Patients who initially sought care from informal providers tended to visit more points before receiving a diagnosis. This increased number of care-seeking points contributes to prolonged diagnostic and health system delays. Consequently, these delays extend the period during which DR-TB can be transmitted within communities. Interrupting this cycle is crucial for the effective control of DR-TB, particularly in resource-limited settings like Bangladesh.

Differentiating the initial symptoms of tuberculosis, which often resemble those of the common cold, COVID-19, or other diseases, is a significant challenge [37]. This difficulty often leads patients to underestimate the seriousness of their symptoms and delay seeking professional healthcare, opting instead to rely on drug sellers. In this study, the median duration from the onset of the first DR-TB symptom to the first healthcare consultation was one week, defined as the patient delay. This is notably shorter than patient delays observed in other countries, such as Zimbabwe (26 days), Ethiopia (35 days), China (58 days in 95% of cases), the USA (25 days), and India (15 days) [32, 34, 38–40]. A contributing factor to the relatively short patient delay in our study is the operational definition used, which considered the time from symptom onset to consultation with any healthcare provider, formal or informal. In contrast, many studies only consider delays to formal healthcare providers [41].

**Table 4. Logistic regression analysis of the determinants of the patient, diagnostic, treatment, and total delay among the DR-TB patients.**

| Variable | Categories | Patient included | Patient delay | | Diagnostic delay | | Treatment delay | | Total Delay | |
|---|---|---|---|---|---|---|---|---|---|---|
| | | | OR | 95% CI | OR | 95% CI | OR | 95% CI | OR | 95% CI |
| **Sex** | Male | 69 | 1 | | 1 | | 1 | | 1 | |
| | Female | 23 | 0.71 | 0.27–1.02 | 1.34 | 0.52–3.46 | 0.44 | 0.17–1.14 | 1.12 | 0.44–2.89 |
| **Age (year)** | 16–34 | 45 | 1 | | 1 | | 1 | | 1 | |
| | 35–54 | 22 | 0.87 | 0.32–2.43 | 1.53 | 0.54–4.37 | 0.96 | 0.34–2.67 | 0.88 | 0.31–2.45 |
| | 55 and above | 25 | 1.31 | 0.49–3.54 | 0.49 | 0.18–1.34 | 1.70 | 0.61–4.74 | 0.34 | 0.12–0.96 |
| **Education** | No education | 44 | 1 | | 1 | | 1 | | 1 | |
| | Primary | 19 | 0.82 | 0.30–2.41 | 1.11 | 0.38–3.26 | 0.46 | 0.15–1.39 | 0.90 | 0.31–2.74 |
| | Secondary & above | 29 | 1.49 | 0.57–3.88 | 1.07 | 0.42–2.74 | 0.64 | 0.24–1.66 | 1.07 | 0.42–2.74 |
| **Residence** | Rural | 66 | 1 | | 1 | | 1 | | 1 | |
| | Urban | 26 | 0.63 | 0.25–1.57 | 0.76 | 0.31–1.89 | 0.49 | 0.19–1.23 | 1 | 0.40–2.48 |
| **Smoke currently** | Yes | 15 | 1 | | 1 | | 1 | | 1 | |
| | No | 77 | 1.05 | 0.35–3.18 | 0.90 | 0.30–2.72 | 0.46 | 0.13–1.57 | 0.62 | 0.53–4.21 |
| **Distance of near health facility** | <5 km | 59 | 1 | | 1 | | 1 | | 1 | |
| | ≥5 km | 33 | 0.57 | 0.24–1.35 | 1.50 | 0.64–3.55 | 0.77 | 0.32–1.82 | 0.91 | 0.39–2.13 |
| **Diabetes** | Yes | 16 | 1 | | 1 | | 1 | | 1 | |
| | No | 76 | 3.19* | 1.01–10.10 | 10.16* | 2.15–47.90 | 2.86 | 0.94–8.71 | 5.65* | 1.49–21.45 |
| **BCG scar** | Yes | 52 | 1 | | 1 | | 1 | | 1 | |
| | No | 40 | 0.73 | 0.32–1.68 | 1.32 | 0.58–3.02 | 1.59 | 0.68–3.72 | 1.43 | 0.62–3.26 |
| **Pre-TB** | Yes | 62 | 1 | | 1 | | 1 | | 1 | |
| | No | 30 | 0.63 | 0.26–1.52 | 0.27 | 0.11–0.69 | 0.39 | 0.16–0.96 | 0.36 | 0.15–0.90 |
| **Complete previous treatment** | Yes | 25 | 1 | | 1 | | 1 | | 1 | |
| | No | 39 | 2.55 | 0.91–7.15 | 1.26 | 0.45–3.48 | 1.13 | 0.39–3.23 | 1.48 | 0.53–4.08 |
| **Number of care-seeking point** | Less than or equal to 4 | 36 | 1 | | 1 | | 1 | | 1 | |
| | More than 4 | 56 | 0.92 | 0.40–2.14 | 3.33* | 1.38–8.03 | 1.49 | 0.64–3.48 | 3.79* | 1.55–9.24 |
| **Type of provider at first point** | Formal | 19 | 1 | | 1 | | 1 | | 1 | |
| | Informal | 73 | 1.09 | 0.39–3.00 | 2.08 | 0.73–5.88 | 1.78 | 0.65–4.94 | 1.14 | 0.41–3.14 |
| **Treatment registration group** | Primary | 63 | 1 | | 1 | | 1 | | 1 | |
| | Secondary | 29 | 1.29 | 0.53–3.13 | 1.92 | 0.78–4.71 | 1.52 | 0.61–3.79 | 1.66 | 0.68–4.04 |
| **Marital status** | Ever married | 76 | 1 | | 1 | | 1 | | 1 | |
| | Never married | 16 | 4.57* | 1.20–17.33 | 1.29 | 0.43–3.81 | 1.21 | 0.40–3.68 | 2.58 | 0.82–8.13 |
| **Consume Alcohol** | Yes | 7 | 1 | | 1 | | 1 | | 1 | |
| | No | 85 | 8.17 | 0.94–70.83 | 0.16 | 0.02–1.35 | 1.07 | 0.23–5.09 | 0.37 | 0.07–2.03 |
| **Consume substances** | Yes | 4 | 1 | | 1 | | 1 | | 1 | |
| | No | 88 | 3.77 | 0.377–37.67 | 0.33 | 0.03–3.33 | 0.46 | 0.05–4.59 | 0.32 | 0.03–3.18 |
| **imprisonment** | Yes | 12 | 1 | | 1 | | 1 | | 1 | |
| | No | 80 | 0.83 | 0.24–2.84 | 1.55 | 0.45–5.29 | 1.02 | 0.30–3.48 | 1 | 0.30–3.36 |
| **Contact with DR-TB** | Yes | 26 | 1 | | 1 | | 1 | | 1 | |
| | No | 56 | 0.36 | 0.13–1.01 | 1.35 | 0.53–3.42 | 0.90 | 0.35–2.33 | 0.92 | 0.36–2.34 |
| | Don't know | 10 | 0.15 | 0.03–0.79 | 1.17 | 0.27–5.02 | 0.63 | 0.14–2.72 | 0.37 | 0.08–1.74 |
| **Asthma** | Yes | 10 | 1 | | 1 | | 1 | 0.25–3.59 | 1 | |
| | No | 82 | 1.21 | 0.33–4.52 | 1.65 | 0.43–6.30 | 0.94 | 0.25–3.59 | 2.57 | 0.62–10.65 |

The healthcare system in Bangladesh, unlike many countries, is pluralistic, with common reliance on drug sellers for initial care. Despite regulations against selling drugs without prescriptions from qualified professionals, monitoring is lax, and the practice persists. In Dhaka, the capital of Bangladesh, two out of five people seek care from community drug sellers [42].

From a patient's perspective, these drug sellers offer proximity, time efficiency, and cost-effectiveness compared to formal healthcare settings.

Drug sellers frequently dispense antibiotics over-the-counter without prescriptions, an illegal but widespread practice in Bangladesh [43]. Their limited knowledge about appropriate antibiotic use [44, 45] contributes to this issue. A recent study revealed that drug sellers in Bangladesh often irrationally prescribe reserve group antibiotics without physician authorization [46, 47]. This misuse of antibiotics can exacerbate antimicrobial resistance. Additionally, pharmaceutical company representatives in South Asian regions, including Bangladesh and Pakistan, unduly influence drug sellers, promoting the unnecessary use of antibiotics [48].

To address these challenges, a strict monitoring and regulatory framework is essential. This should target pharmaceutical industries to ensure ethical marketing practices, discourage collaboration with informal providers, and emphasize more on research and development.

Our study uncovered longer diagnostic delays among participants than anticipated, diverging from previous findings. A study conducted in Bangladesh during 2012–13 among MDR-TB patients reported a health system delay of 7.1 weeks and a provider delay of 4 weeks [10], considerably shorter than our findings. The difference may be partly attributed to the role of informal providers. In the previous study, a mere 10.8% of participants initially sought care from informal providers, in stark contrast to 81.32% in our study. Additionally, our study had a higher proportion of rural participants (71.74% vs. 49.3%) and illiterates (45.65% vs. 22%) compared to the study by Rifat et al. Furthermore, 27.17% of our study participants were new MDR-TB cases, whereas the earlier study reported only 2.4% [49].

In comparison with other countries, the median diagnostic delay among MDR-TB patients in China and Zimbabwe was 84 and 97 days, respectively [34, 38, 50], aligning closely with our findings. Another study in Bangladesh reported a diagnostic delay of 68.5 days among TB patients [15], shorter than in our study. Factors contributing to diagnostic delay include initial care-seeking from informal providers, consulting multiple healthcare providers, and frequent switching between providers. Moreover, even qualified healthcare providers sometimes fail to recommend the necessary investigations for diagnosing DR-TB [51].

Diagnostic delay is a critical indicator of health system performance. Addressing and reducing these delays, particularly in the context of Bangladesh's pluralistic health system, remains a significant challenge.

Our study identified shorter median treatment initiation delays compared to several previous studies. Earlier research reported median treatment initiation delays of 9 days in China [52], 8 days in Gujarat, India [53], 13 days in Myanmar [54], 10 days in Bangladesh [55], 10–22 days in South Africa [56], 9–17 days in Russia [57], and 7 days in Bhopal, India [58]. Common causes of these delays include the need for repeat confirmatory tests, baseline investigations before commencing treatment, and occasionally, a shortage of hospital beds designated for DR-TB patients in tertiary care facilities [55]. Additionally, patient factors such as a lack of awareness about the disease and motivation to seek timely treatment can contribute to these delays [59].

Treatment delays can lead to a poor prognosis and increase the risk of DR-TB transmission within the community. To mitigate these delays, strategies such as ensuring an adequate number of DR-TB beds in all tertiary hospitals, providing effective counseling post-diagnosis, and facilitating timely baseline investigations upon the admission of DR-TB patients can be implemented.

The median total delay identified in our study was longer compared to some previous studies conducted in Bangladesh and other countries. Previous research found median diagnostic delays of 57 days in the USA [39], 120 days in China, 100 days in Pakistan [27], and 132 days in Zimbabwe. In Bangladesh, Rifat et al. reported a median health system delay of 7.1 weeks

among MDR-TB patients in 2015 [55]. Another study among TB patients in Bangladesh identified a total delay of 12 weeks [35]. The cumulative impact of patient delays, diagnostic delays, and treatment delays contributes to the overall length of total delays. Both patient behaviors and health system inefficiencies play roles in these prolonged delays.

During these extended periods, resistant tuberculosis bacilli continue to spread from one individual to another. The patient's lung condition may deteriorate, leading to poorer disease prognosis, increased risk of treatment failure, higher mortality rates, and substantial economic losses. The risk of fatality escalates significantly with longer total delays [55]. Addressing these delays is crucial for improving patient outcomes and controlling the spread of DR-TB.

Our study revealed that the median patient delay for female participants was nearly double that of their male counterparts. Additionally, females experienced longer diagnostic, treatment, and total delays compared to males. A significant proportion of female participants were housewives, who also exhibited longer delays across various categories of occupation. Previous research has highlighted the inability to make decisions independently as a crucial factor contributing to patient delays [60, 61]. In Bangladesh's typically male-dominated family structures, where men are often the primary decision-makers [62, 63], this trend contributes to longer delays in healthcare-seeking among women, a finding consistent with earlier studies [63, 64].

Rural communities were found to be particularly vulnerable to DR-TB, facing a threefold higher risk of developing MDR-TB compared to urban populations [60]. Our study observed that rural participants had longer diagnostic, treatment, and total delays, which is concerning. Factors such as financial constraints, social stigma, limited knowledge about TB, lack of awareness, and transportation challenges significantly influence healthcare-seeking delays among rural residents. Participants from lower-income groups tended to experience longer diagnostic delays, as financial resources are essential for accessing investigations, transportation, and other healthcare facilities [60]. Our findings align with this observation, indicating longer diagnostic delays among less affluent participants.

Given these disparities, policymakers and health professionals should prioritize interventions targeting women and rural communities. Special attention to these subgroups is vital to reduce healthcare-seeking delays and mitigate the spread and impact of DR-TB.

In our study, being non-diabetic was significantly associated with longer patient, diagnostic, and total delays. In Bangladesh, a majority (over 50%) of diabetic patients are registered with the Bangladesh Diabetic Somity (BADAS) and predominantly seek care (95.2%) from BADAS's private facilities, while only a small fraction (4.3%) turn to informal providers [65]. This organized care-seeking pattern among diabetic patients likely contributes to their shorter delays compared to non-diabetic individuals.

Additionally, we found that never being married was significantly associated with longer patient delays. Unmarried individuals may lack self-motivation and social support, which could explain the extended patient delays observed in this group [66]. Furthermore, consulting more than four care-seeking points was significantly associated with increased diagnostic and total delays. It is intuitive that consulting a greater number of healthcare points leads to prolonged delays in the diagnosis and treatment of DR-TB.

This study's primary strength lies in the face-to-face interviews conducted with DR-TB patients, enabling direct validation of delay-related information against patient recollections and medical records. Despite the lower case detection rate of DR-TB in Bangladesh [67, 68], our study successfully encompassed a diverse range of patients from both rural and urban areas, and included individuals treated in both government and non-government TB hospitals across specific geographical regions.

However, the study was not without its limitations. One potential issue was recall bias regarding patient delays and healthcare-seeking behaviors, as the data collection relied heavily on patient memory. To mitigate this, most participants were selected from the intensive phase of treatment, thus reducing the recall period compared to community patients. Another challenge involved incomplete medical records, which occasionally made it difficult to ascertain precise dates of diagnosis or treatment initiation. Efforts were made to minimize these discrepancies by thoroughly reviewing all available medical and hospital service records.

Additionally, some data on patient delays were missing, as a few patients could not recall the exact date of their initial healthcare consultation. Consequently, calculating total delays for these individuals was not feasible. Nevertheless, this missing data did not significantly impact the overall outcomes of the study, ensuring the findings remained robust and representative of the study population.

The study underscores the need for enhancing societal awareness to reduce reliance on informal healthcare providers. To this end, stringent monitoring and regulatory controls are essential to prevent over-the-counter medication prescribing by drug sellers. Establishing a robust referral system is crucial, one that efficiently facilitates patient transfer among private and public healthcare facilities, from informal to formal healthcare providers, and across different levels of care, from primary to tertiary facilities.

To further diminish health system delays and bridge the existing gaps in the healthcare infrastructure, policymakers must take decisive actions. These include increasing financial investment in the health sector, strengthening the overall healthcare system with a focus on enhancing both diagnostic and treatment capacities, and providing comprehensive training to healthcare personnel. These measures are vital for the effective management of DR-TB and for improving patient outcomes in the long term.

## Conclusion

The study highlights that a majority of DR-TB patients initially sought care from informal healthcare providers, and often visited more than five different care points before receiving a diagnosis, leading to extended patient and diagnostic delays. These delays, including the delay in treatment initiation, were found to be longer than recommended. Consequently, this contributes to the community transmission of drug-resistant TB bacilli and exacerbates the disease burden.

To address these issues, community-based interventions are essential. These should focus on guiding patients towards formal healthcare providers and reducing the number of consultations before achieving a diagnosis. Enhancing training and awareness among healthcare providers could significantly curtail diagnostic delays. Furthermore, the study advocates for the widespread availability of rapid diagnostic tests, such as GeneXpert, at primary healthcare centers.

The insights gained from this research can aid policymakers, public health specialists, and healthcare providers in refining TB control programs. By promoting public awareness to minimize patient delays, strengthening health systems to decrease diagnostic delays, and ensuring prompt hospital admissions to reduce treatment delays, these efforts align with the overarching goal of successfully implementing the 'End TB 2030' strategy.

## Supporting information

**S1 Checklist. Inclusivity in global research.**
(DOCX)

## Acknowledgments

We extend our profound gratitude to Dr. Aung Kya Jai Maug, Country Director of the Damien Foundation, and Dr. Abdul Hamid Selim, Senior Scientist at the National Tuberculosis and Leprosy Control Programme under the Ministry of Health and Family Welfare, Government of the People's Republic of Bangladesh. Their unwavering support and cooperation were instrumental throughout the duration of this study. Additionally, our sincere thanks go to Avijit Saha, Ria Azmi, and Mushfiqur Rahman, who served as mentors for the summative learning project of which this study is a part. Their guidance and insights have been invaluable in the successful completion of this research.

## Author Contributions

**Conceptualization:** Md. Zulqarnine Ibne Noman, Shaki Aktar, Ateeb Ahmad Parray, Dennis G. Amando, Jyoti Karki, Zafria Atsna, Dipak Kumar Mitra, Shaikh A. Shahed Hossain.

**Data curation:** Md. Zulqarnine Ibne Noman, Shaki Aktar, Dennis G. Amando, Jyoti Karki, Zafria Atsna, Dipak Kumar Mitra, Shaikh A. Shahed Hossain.

**Formal analysis:** Md. Zulqarnine Ibne Noman, Shariful Islam, Dennis G. Amando, Zafria Atsna, Dipak Kumar Mitra, Shaikh A. Shahed Hossain.

**Investigation:** Md. Zulqarnine Ibne Noman, Shaki Aktar, Ateeb Ahmad Parray, Dennis G. Amando, Jyoti Karki, Zafria Atsna, Dipak Kumar Mitra, Shaikh A. Shahed Hossain.

**Methodology:** Md. Zulqarnine Ibne Noman, Shariful Islam, Shaki Aktar, Ateeb Ahmad Parray, Dennis G. Amando, Zafria Atsna, Dipak Kumar Mitra, Shaikh A. Shahed Hossain.

**Project administration:** Md. Zulqarnine Ibne Noman, Ateeb Ahmad Parray, Dennis G. Amando, Jyoti Karki, Zafria Atsna, Dipak Kumar Mitra, Shaikh A. Shahed Hossain.

**Resources:** Md. Zulqarnine Ibne Noman, Shaki Aktar, Ateeb Ahmad Parray, Dennis G. Amando, Jyoti Karki, Zafria Atsna, Dipak Kumar Mitra, Shaikh A. Shahed Hossain.

**Software:** Md. Zulqarnine Ibne Noman, Shaikh A. Shahed Hossain.

**Supervision:** Dennis G. Amando, Zafria Atsna, Dipak Kumar Mitra, Shaikh A. Shahed Hossain.

**Validation:** Md. Zulqarnine Ibne Noman, Shariful Islam, Shaki Aktar, Ateeb Ahmad Parray, Dennis G. Amando, Jyoti Karki, Zafria Atsna, Dipak Kumar Mitra, Shaikh A. Shahed Hossain.

**Visualization:** Md. Zulqarnine Ibne Noman, Shariful Islam, Ateeb Ahmad Parray, Shaikh A. Shahed Hossain.

**Writing – original draft:** Md. Zulqarnine Ibne Noman, Shaikh A. Shahed Hossain.

**Writing – review & editing:** Md. Zulqarnine Ibne Noman, Shariful Islam, Shaki Aktar, Ateeb Ahmad Parray, Dennis G. Amando, Jyoti Karki, Zafria Atsna, Dipak Kumar Mitra, Shaikh A. Shahed Hossain.

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
