## [Decision Letter · Decision Letter 0]

26 May 2023

PGPH-D-23-00641

Healthcare seeking behavior and delays in case of Drug-Resistant Tuberculosis patients in Bangladesh: Findings from a cross-sectional survey

Dear Dr. Parray,

Thank you for submitting your manuscript to PLOS Global Public Health. After careful consideration, we feel that it has merit but does not fully meet PLOS Global Public Health’s publication criteria as it currently stands. Therefore, we invite you to submit a revised version of the manuscript that addresses the points raised during the review process.

We look forward to receiving your revised manuscript.

Kind regards,

Sadia Shakoor

Academic Editor

Journal Requirements:

2. Some material included in your submission may be copyrighted. According to PLOS’s copyright policy, authors who use figures or other material (e.g., graphics, clipart, maps) from another author or copyright holder must demonstrate or obtain permission to publish this material under the Creative Commons Attribution 4.0 International (CC BY 4.0) License used by PLOS journals. Please closely review the details of PLOS’s copyright requirements here: PLOS Licenses and Copyright. If you need to request permissions from a copyright holder, you may use PLOS's Copyright Content Permission form.

Potential Copyright Issues:

Figure 1: please (a) provide a direct link to the base layer of the map (i.e., the country or region border shape) and ensure this is also included in the figure legend; and (b) provide a link to the terms of use / license information for the base layer image or shapefile. We cannot publish proprietary or copyrighted maps (e.g. Google Maps, Mapquest) and the terms of use for your map base layer must be compatible with our CC-BY 4.0 license. 

Reviewers' comments:

Reviewer's Responses to Questions

**Comments to the Author**

1. Does this manuscript meet PLOS Global Public Health’s publication criteria? Is the manuscript technically sound, and do the data support the conclusions? The manuscript must describe methodologically and ethically rigorous research with conclusions that are appropriately drawn based on the data presented.

Reviewer #1: Yes

Reviewer #2: Yes

2. Has the statistical analysis been performed appropriately and rigorously?

Reviewer #1: Yes

Reviewer #2: Yes

3. Have the authors made all data underlying the findings in their manuscript fully available (please refer to the Data Availability Statement at the start of the manuscript PDF file)?

Reviewer #1: Yes

Reviewer #2: Yes

4. Is the manuscript presented in an intelligible fashion and written in standard English?

Reviewer #1: Yes

Reviewer #2: Yes

5. Review Comments to the Author

Reviewer #1: 

Introduction: The introduction needs refinement in both sense, scientifically and grammatically. Drug Resistant section definitions should be updated like MDR-TB, XDR-TB latest definitions should be incorporated.

Methodology: Operational section in methodology should be revised (like definitions of new, re-treatment and relapse cases).

Results:

Result section should be reviewed as mentioned in text majority of the participants are male while in table-3 it’s mentioned as female.The citations of tables/figures, should be uniform.Results discussed should be same both in text, tables and figures.

Discussion:

Include literature review from recent studies about the topic if available.

Reviewer #2: Thank you for allowing me to review the manuscript titled “Healthcare seeking behaviour and delay in case of Drug-Resistant Tuberculosis patients in Bangladesh: Findings from a cross-sectional survey”. I found the analysis presented in this paper very important. Not only, this paper is a contribution to the body of knowledge on antimicrobial resistance research, but it certainly has important implications for policy and practice in Bangladesh. I recommend this paper for publication, but there are several areas on which the authors need to do some further work. My suggested revisions intend to bring clarity to the writing and the data in several parts of the paper. 

Introduction:

- The authors talk about the End-TB 2030 strategy, but it is not clear where this strategy comes from. Please briefly give details about it.

- On page 5, from lines 98-99 to me, healthcare-seeking behaviour is not an influential factor. As indicated, it is a behaviour/tendency about accessing and utilising healthcare in response to given ailments, and it is typically shaped by the contexts in which human beings live (i.e., socioeconomic circumstances, beliefs about health, stigma, accessibility). The statement, therefore, needs to be conceptually correct.

- I noticed several terms have been interchangeably used to refer to the same thing; for instance, what is the difference between drug sellers/pharmacists OR traditional healers/traditional providers/informal providers/ayurvedic practitioners OR village doctors/doctors? Please be specific and consistent about using these terms, as a lack of clarity in them can confuse the reader in understanding the analysis.

Methods:

- Structured questionnaires are tools that help collect data about various variables. They do not help to conduct “face-to-face interviews”. The term face-to-face interviews transmit a message about a qualitative design, which was not used in the study. Maybe you write as “a pretested structured questionnaire was used to gather quantitative data OR data about XYZ”.

Results:

- Page 12, line 242: “Most of the patients 42 (45.65%) had never attended school”. 45% is “less than half” of the total sample, and it should be written accordingly. Please make changes to other similar descriptions, with appropriate classifications: few (2-5), a few (6-15), some (16-30), many (31-39), less than half, more than half, and most/majority of etc – just to give an example to come up with criteria to consistently describe the numbers. Logical descriptions are very important when the quantitative data is explained. Please look at all the numbers logically/objectively.

- How did you ask about “Haemoptysis” and “anorexia” from patients? I suspect that this is the authors’ conceptualisation of the symptoms reported by the patients. Perhaps, simplify them as asked via the questionnaire or told by the patients (cough with blood, for instance).

Discussion:

This section typically allows authors to reflect on findings and tell the readers what is meant by what they have found (in the results section). However, most of the content presented in the section is a simple restatement of the findings discussed earlier. It is always useful to restate findings, but limiting it to the initial first paragraph is enough. This section must include some further analysis of why certain segments of the sample were more likely to experience delays than others in terms of seeking care, diagnosis, and treatment. I am aware that the authors talked about factors like informal providers and over-the-counter medication. But there is a wealth of literature in low- and middle-income countries such as India and Pakistan speaking about informal providers’ links with pharmaceutical companies and how to achieve targets, informal providers overprescribe antibiotics. This literature (from Bangladesh and other countries) needs to be included in the analysis.

Additionally, there is not much focus on the study findings’ implications for policy and practice. The authors need to describe how they see the research can help identify gaps in Bangladesh’s health system (regulation, policy, and practice), and what things could be improved within the health system to address the issue of delayed diagnosis and treatment of TB. Also, now that we know that in Bangladesh DR-TB is prevalent, and also some factors are contributing to it, can we speak something about the pharmaceutical industry and how could it play a part to address the issue (i.e., promoting ethical prescribing, refraining to work with informal providers, and focusing more on R&D)?

6. PLOS authors have the option to publish the peer review history of their article (what does this mean?). If published, this will include your full peer review and any attached files.

**Do you want your identity to be public for this peer review?** For information about this choice, including consent withdrawal, please see our Privacy Policy.

Reviewer #1: No

Reviewer #2: **Yes: **Muhammad Naveed Noor

---

## [Editor Report · Decision Letter 1]

4 Oct 2023

Healthcare seeking behavior and delays in case of Drug-Resistant Tuberculosis patients in Bangladesh: Findings from a cross-sectional survey

PGPH-D-23-00641R1

Dear Mr. Parray,

We are pleased to inform you that your manuscript 'Healthcare seeking behavior and delays in case of Drug-Resistant Tuberculosis patients in Bangladesh: Findings from a cross-sectional survey' has been provisionally accepted for publication in PLOS Global Public Health.

Best regards,

Sadia Shakoor

Academic Editor